# On the Use of a Simplified Slip Limit Equation to Predict Screw Self-Loosening of Dental Implants Subjected to External Cycling Loading

**Mikel Armentia** [1,2,*]**, Mikel Abasolo** [1]**, Ibai Coria** [1]  **and Abdel-Hakim Bouzid** [3]

[1] Mechanical Engineering Department, University of the Basque Country, 48013 Bilbao, Spain; mikel.abasolo@ehu.eus (M.A.); ibai.coria@ehu.eus (I.C.)

[2] R&D Department, Biotechnology Institute I mas D S.L., 01510 Miñano, Spain

[3] Mechanical Engineering Department, École de Technologie Supérieure, Montreal, QC H3C 1K3, Canada; hakim.bouzid@etsmtl.ca

[*] Correspondence: mikel_armentia@hotmail.com

**Featured Application: A methodology to study screw self-loosening phenomenon in preliminary design stages of dental implants is presented.**

**Abstract:** Self-loosening of the prosthetic screws is a major mechanical problem affecting roughly 10% of dental implants, according to the literature. This phenomenon may lead to micro-movements that produce crestal bone loss, peri-implantitis, or structural failure of the implant assembly. In this paper, a simple and effective tool to predict self-loosening under masticatory loads is presented. The loads acting on the screw are obtained from a simple finite element (FE) model, and introduced in a mathematical formula that calculates the torque needed to loosen the screw; self-loosening will occur when this torque becomes zero. In this sense, all the parameters involved in self-loosening phenomenon can be easily identified, and their effect quantified. For validating purposes, 90 experimental tests were performed in a direct stress test bench. As a result, a powerful tool with a maximum experimental error of 7.6% is presented, allowing dental implant manufacturers to predict eventual occurrence of self-loosening in their developed dental implant products and take corrective actions at preliminary design stage. Furthermore, the following clinical implications can be directly derived from the methodology: a higher screw preload, that is a higher tightening torque, improves self-loosening response of the dental implant and, similarly, for a given preload force, higher friction coefficient and screw metric, as well as lower pitch and thread angle values, are also found to be beneficial.

**Keywords:** dental implant; self-loosening; analytical tool; design methodology; mechanical approach

## 1. Introduction

In dental implant restorations, implant and prosthetic elements are usually held together by means of a screw [1]. A tightening torque is applied to the screw head in order to create a tension load known as preload [2] and ensure a good structural integrity among the components [3,4]. The recommended torque is provided by manufacturers based on different implant design factors [5].

Nevertheless, during their life span dental implants are subjected to variable loads; biting, mastication and bruxism loads to name a few. Axial loads are generally predominant but are always accompanied by lateral forces [6]. These loads can cause the screw to untighten or loose preload; this is known as screw self-loosening phenomenon and is produced by the lateral external force that generate a rotational movement of the screw. As a result, gradual preload loss may occur, leading to micro-movements and eventual structural failure of the dental implant assembly [2,7–11].

Screw loosening is one of the major mechanical cause of tooth implant replacement [12], affecting between 4.3% and 12.7% of dental implants according to specialized literature [13–15]. It is well established that screw loosening depends on a large number of parameters, such as tightening torque, preload, occlusal forces, thread embedment and geometrical misfits [10,16,17]. In this sense, screw loosening is generally the result of an inadequate tightening torque, incorrect implant design, manufacturing defects or unexpected load conditions, amongst others [18–20].

Many works have studied the influence of different parameters of screw self-loosening. Wu et al. concluded that lubrication reduces friction and consequently increases the clamping force while reducing the loosening torque [21]. Elias et al. studied the effect of screw coatings that decreases the friction coefficient, agreeing with Wu et al. in terms of preload and untightening torque [22]. Park et al. recommend the clinical use of gold-plated screws to prevent screw loosening. Teflon coating is also recommended by these authors as well as avoiding repeated tightening [23]. In several works, a significant reduction of removal torque after cyclic loading was noticed, demonstrating that screw preload decreases with the number of cycles due to screw self-loosening [9,24]. Siamos et al. studied the influence of the screw tightening torque and consequently the screw preload level and recommended a tightening torque above 30 Ncm to minimize screw self-loosening [25]. In this sense, Lang et al. stated that the preload should be 75% of the yield strength of the screw [26]. Aboyoussef et al. concluded that anti-rotation strategies may be used to reduce screw self-loosening problems [16]. Along this line, Arshad et al. proposed adding an adhesive to the screw joint interface [19]. The effect of other parameters on the screw self-loosening, such as abutment type, has also been studied in dental implant applications [27].

Researchers agree that transverse loading is a major source of self-loosening in bolted joints in general [28–33] and in dental implants in particular [6,9,16,24]. In this sense, Nassar et al. [31,32] presented a mathematical model to estimate self-loosening under transverse cyclic loads, and validated it using an experimental setup with two sliding plates. Fort et al. [33] found some limitations in these mathematical model such as the difficulty to analytically obtain the bending stiffness under the bolt head and nut, and further developed a model that includes the effect of plate thickness on self-loosening. In general, this model is very useful to understand the mechanical foundations of the phenomenon and the effect of different design parameters. Some conclusions were derived from this work; the thickness of the clamping parts increases self-loosening and so does the amplitude of the transverse vibrations. Meanwhile, higher friction coefficient, lower modulus of elasticity and lower thread pitch value improve self-loosening response. As a drawback, the model must be solved using numerical integration rendering its applicability not straightforward to use. The present work simplifies the models in [31,32] to a simple equation for use in a step-by-step methodology to predict screw self-loosening of dental implants. Due to its simple and intuitive nature, the methodology not only allows a better understanding of the self-loosening phenomenon and the parameters involved in its evolution in dental implants, but its implementation in a tool help manufacturers make suitable and efficient decisions during the process chain of design, manufacturing and assembly.

## 2. Materials and Methods

### 2.1. Background: Loosening Torque

Figure 1 shows the free body diagrams of a screw during tightening and untightening operations showing the different torques involved. $T_T$ and $T_L$ are, respectively, the applied external tightening and untightening (loosening) torques. $T_h$ and $T_t$ are the resistance torques developed under the screw head and in the thread contact surface due to friction forces, respectively, both acting in the opposite direction to the screw rotation. Finally, $T_p$ is the pitch torque generated by the axial load acting on the helical surface of the threads, always acting in the screw loosening direction. Thus, the external tightening and loosening torques $T_T$ and $T_L$ can be expressed in terms of the following twisting moment equilibrium equations:

$$T_T = T_h + T_t + T_p \tag{1}$$

$$T_L = T_h + T_t - T_p \tag{2}$$

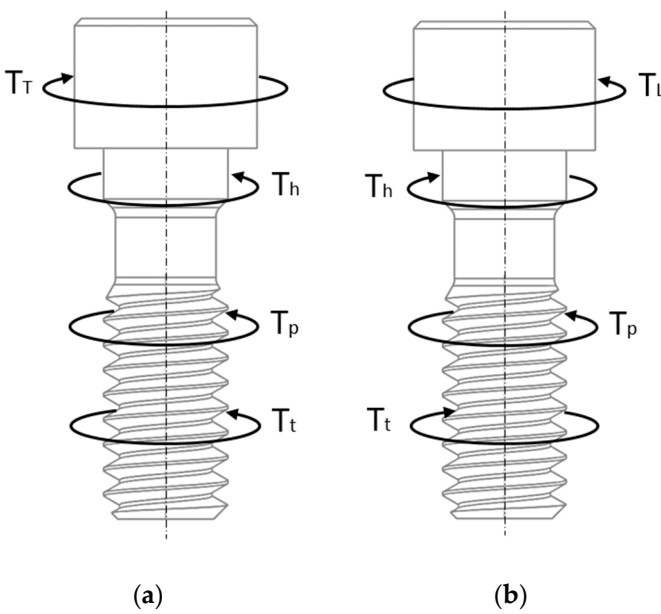

(**a**)          (**b**)

**Figure 1.** Twisting moment equilibrium during: (**a**) tightening (**b**) untightening.

Based on the Motosh methodology and including the 3D effect of both the helix and thread profile angles the approximate torque tension relationship is to give [34,35]:

$$T_T = F_p \cdot \mu_h \cdot r_h + F_p \cdot \mu_t \cdot \frac{r_t}{\gamma \cdot cos\alpha} + F_p \cdot \frac{tan\beta}{\gamma} \cdot r_t \tag{3}$$

$$T_L = F_p \cdot \mu_h \cdot r_h + F_p \cdot \mu_t \cdot \frac{r_t}{\gamma \cdot cos\alpha} - F_p \cdot \frac{tan\beta}{\gamma} \cdot r_t \tag{4}$$

where the first term is $T_h$, the second one $T_t$ and the third one $T_p$. The more accurate expressions for (3), (4) are given in [32]. In Equations (3) and (4), $F_p$ is the screw preload, and $\mu_h$ and $\mu_t$ are respectively the friction coefficients of the screw head and thread contacts. Besides, as illustrated in Figure 2, $\alpha$ is the half-angle of the thread profile, $\beta$ is the helix angle, and $r_h$ and $r_t$ are respectively the screw head and thread effective contact radii obtained by considering the preload to be uniformly distributed under the bolt head and on the thread surface:

$$r_h = \frac{2}{3} \cdot \frac{r_{hmax}^3 - r_{hmin}^3}{r_{hmax}^2 - r_{hmin}^2} \approx \frac{r_{hmax} + r_{hmin}}{2} \tag{5}$$

$$r_t = \frac{2}{3} \cdot \frac{r_{tmax}^3 - r_{tmin}^3}{r_{tmax}^2 - r_{tmin}^2} \approx \frac{r_{tmax} + r_{tmin}}{2} \tag{6}$$

where $r_{hmax}$ and $r_{hmin}$ are the maximum and minimum radius of the screw head contact surface, and $r_{tmax}$ and $r_{tmin}$ are the maximum and minimum radius of the screw thread contact surface (see Figure 2). Finally, the parameter $\gamma$ is:

$$\gamma = 1 - \mu_t \cdot \frac{sin\beta}{cos\alpha} \tag{7}$$

Thus, Equation (3) allows putting into relation the applied tightening torque and the preload achieved in the screw, and similarly with the loosening torque (Equation (4)). Nevertheless, Equations (3) and (4) do not consider the effect of an external transverse force on the tightening and loosening

torques. As mentioned in the Introduction section, this force contributes to overcome the resisting friction torques. Consequently, the loosening torque needed to untighten the screw when an external transverse force is acting on it will be smaller than the value predicted in (4).

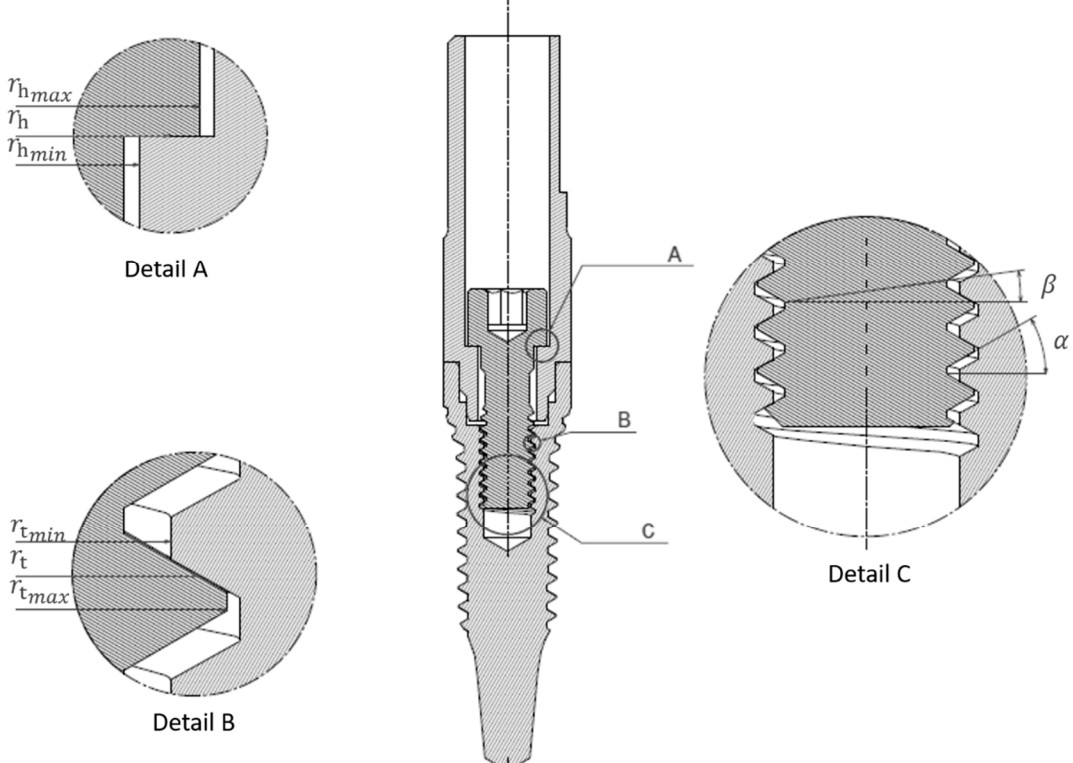

**Figure 2.** Geometric parameters of the screwed joint of a dental implant assembly.

In this sense, Nassar et al. [31,32] developed an analytical model to study self-loosening in screwed joints under transverse cyclic loading, which experimentally proved to give accurate results for the case of two sliding plates, similar to Junker test machine [28,29]. The effect of the screw head flexibility was further implemented in the model by Fort et al. [33]. This model allows understanding the mechanical foundations of the phenomenon and quantifying the effect of the design, manufacturing and operational parameters involved, thus enabling to take the correct decisions in terms of screw head and thread geometry, friction coefficient, tightening torque, and other relevant parameters. As a drawback, its applicability is not straightforward, as the analytical model must be solved via numerical integration.

*2.2. Simplified Model for the Loosening Torque*

Based on these works by Nassar et al. [31,32] and Fort et al. [33], the present work proposes a simpler analytical formula, obtained by curve-fitting, to predict self-loosening. The complete development of the model is detailed in the Supplementary Materials section. As a final result, the following equation that expresses the loosening torque $T_L$ and therefore predicts self-loosening in a simple and straightforward way was obtained:

$$T_L = \mu_h \cdot F_a \cdot r_h \cdot \langle 1 - \frac{F_e}{\mu_h \cdot F_a} \rangle^{1/2} + \mu_t \cdot F_a \cdot \frac{r_t}{\gamma \cdot cos\alpha} \cdot \langle 1 - \gamma \cdot \delta \cdot \frac{F_e}{\mu_t \cdot F_a} \rangle^{1/2} - \left( F_a \cdot \frac{tan\beta}{\gamma} \cdot r_t \right) \tag{8}$$

Being:

$$\delta = \frac{1 + cos\alpha}{2} \tag{9}$$

where, $F_e$ is the transverse load acting on the screw and $F_a$ is its axial load. Note that $F_a$ is not necessary equal to $F_p$ in (3)–(4), since the force that introduces a transverse force $F_e$ may also change the axial force in the screw, making it different to $F_p$. Additionally, Equation (8) can be considered as a generalization of the classical torque-preload equation with the first term being $T_h$, the second one $T_t$ and the third one $T_p$, when the transverse load $F_e$ is set to zero. As $F_e$ helps reducing the loosening torque, this variable appears in the first and second terms reducing the underhead and thread frictional torques. In this sense, an important observation must be made regarding $T_h$ and $T_t$: once any of these two resisting torques is completely overcome, i.e., once the expression between Macaulay brackets becomes negative, the term is put to zero in Equation (8).

Equation (8) predicts the screw self-loosening for a combination of an axial and transverse load $F_e$ that makes the loosening torque $T_L$ becomes zero. Its main advantage is its torque-preload-like form, which makes it quite simple to use and intuitive to understandable while it quantifies the influence of the parameters involved in the phenomenon of self-loosening. As a particular application, the model has been successfully implemented in a new step-by-step methodology to study and predict screw self-loosening in dental implants that is presented in the following section.

### 2.3. Methodology to Study Self-Loosening in Dental Implants

The flowchart in Figure 3 shows a four-step methodology. In the first step, input data is entered namely the dental implant geometry, friction coefficients, screw tightening torque $T_T$ and, finally, the external masticatory load $F$ which is applied on the abutment. In the second step, the preload value $F_p$ is calculated from the tightening torque $T_T$ according to Equation (3). In the third step, a finite element (FE) model of the dental implant with the geometry and friction coefficient values of Step 1 must be analyzed; first, the screw is preloaded to the preload value $F_p$ calculated in Step 2, and afterwards the masticatory load $F$ is applied to the abutment; the outputs of the FE model are the transverse and axial loads $F_e$ and $F_a$ acting on the screw head-abutment contact, i.e., the transverse and axial contact reactions acting on the screw head contact surface. It is to be noted that, as mentioned before, $F_a$ may be different to $F_p$, because both the axial component of the applied force $F$ on the abutment and its transverse component that create bending may alter the axial force of the screw. In the fourth and last step of the methodology, $F_e$ and $F_a$ obtained from the FE model are introduced in Equation (8). If $T_L > 0$, the dental implant under study, assembled with the tightening torque $T_T$ and subjected to the masticatory load $F$ will not suffer screw self-loosening. Otherwise, corrective actions will be necessary to avoid self-loosening such as a change of one or a combination of the following parameters; the implant global geometry, the screw thread configuration, the friction coefficients and the screw tightening torque. Thus, the flowchart in Figure 3 can be implemented in the iterative design process of dental implants to improve self-loosening performance.

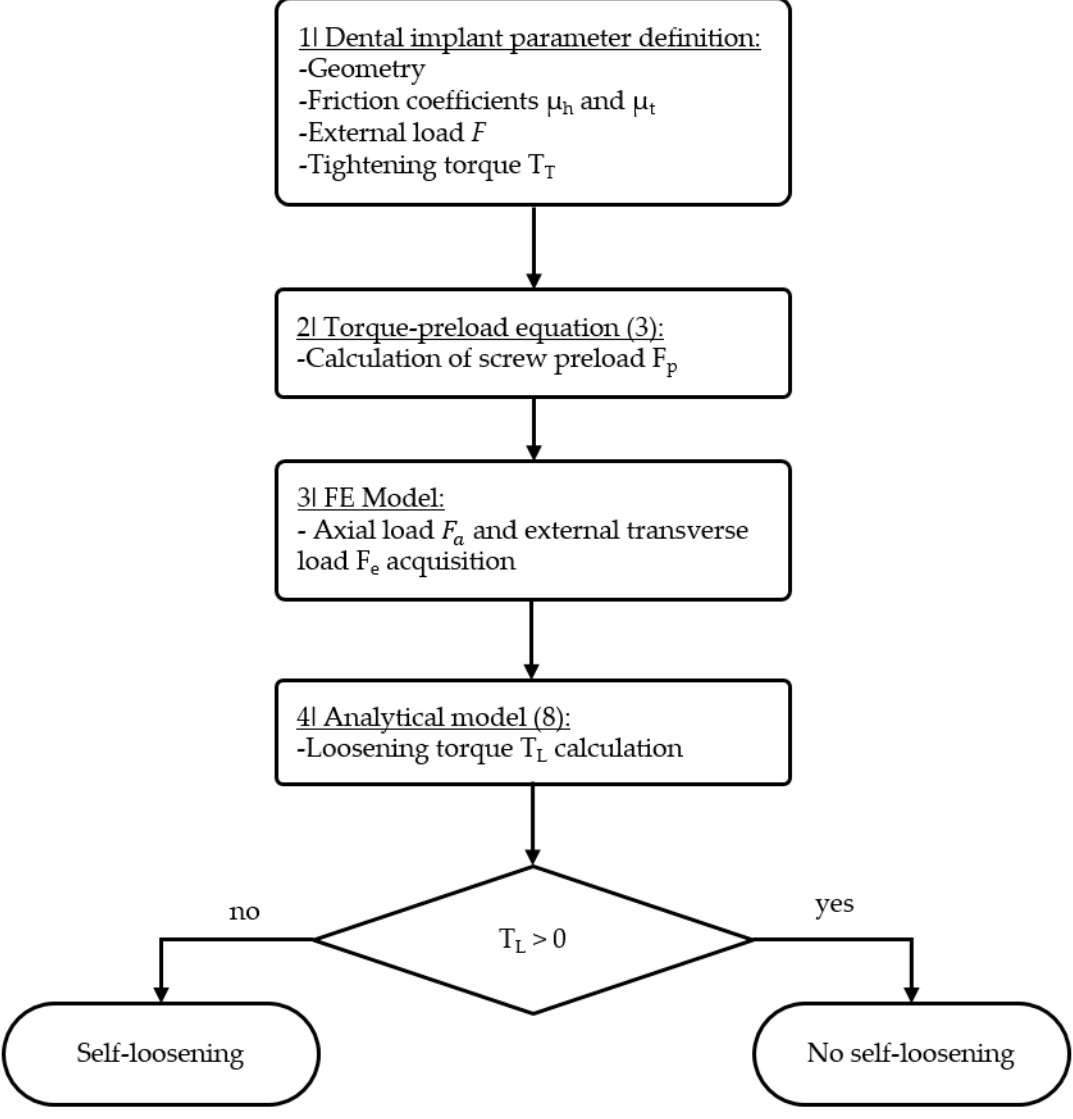

**Figure 3.** Methodology to estimate screw self-loosening in dental implants.

*2.4. Experimental Setup*

Experimental tests were performed on a dental implant model provided by BTI (Biotechnology Institute, Vitoria, Spain). The purpose of these tests was to validate the methodology presented in Figure 3. The dental implant assembly under study is composed by a BTI INTERNA® IIPUCA3313 implant with a 3.3 mm body diameter and a 4.1 mm Universal Platform with a four-lobe anti-rotation connection, a BTI INPPTU44 titanium abutment (used for direct restorations) and an INTTUH retaining screw with TiBlack® coating (see Figure 4). The half-angle of the thread profile is $\alpha = 30°$ and the helix angle is $\beta = 5.75°$ (0.35 mm pitch). The screw head and thread effective radii are $r_h = 1.115$ mm and $r_t = 0.810$ mm. The friction coefficients are $\mu_t = \mu_h = 0.17$ for the screw contacts and 0.21 for the implant-abutment contact, as measured in a MicroTest SMT-A/0200 pin-on-disk tribometer (Microtest S.A., Madrid, Spain) in a previous work by the authors (see Figure 5) [36]. The experimental tests were carried out by applying a cyclic load on the implant by means of an INSTRON 8801 servo-hydraulic direct stress test bench (Instron, Barcelona, Spain). Force control was used to apply the load cycles using a DYNACELL™ 2527-129 load cell (±2 kN load range). The dental implant was attached to the specimen holder hole by using Loctite 401, which is an embedding material that meets the specifications of [37].

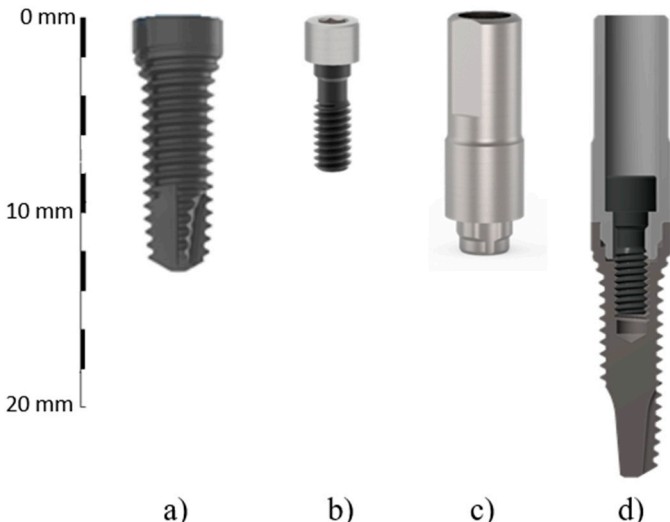

**Figure 4.** Dental implant under study: (**a**) implant, (**b**) screw, (**c**) abutment, and (**d**) assembly.

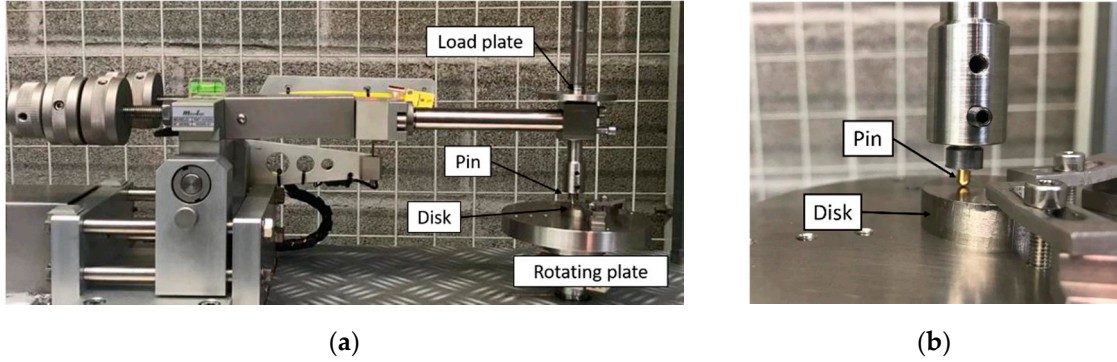

**Figure 5.** Pin-on-disk tribometer: (**a**) general view, (**b**) detail view [36].

Using the predictions obtained from the methodology of Figure 3, the preliminary experimental tests confirmed that, under moderate external load F applied at 30° [37], the tightening torque recommended by the manufacturer (35 Ncm) is sufficient to avoid self-loosening. Extreme values of *F* that may structurally harm the dental implant are not of interest. As the aim of this work is to present and validate the screw self-loosening methodology, some modifications were introduced in order to promote the self-loosening phenomenon in the dental implant under study. Firstly, the screw tightening torque was reduced to 10, 15 and 20 Ncm. Secondly, the thickness of the lower part of the abutment was reduced 0.19 mm in diameter to allow bigger lateral movements of the abutment (see Figure 6). Finally, the cyclic external load *F* was applied in both directions (pure alternating sinusoidal force) at a frequency of 1Hz on the bottom of the abutment perpendicular to the axis of the dental implant (i.e., with an inclination of 90°). For this purpose, a special loading jig was designed and manufactured to hold the dental implant and transmit the cyclic external load *F* to the bottom of the abutment as shown in Figure 7. A steel ring is installed around the abutment ouside surface near the junction with the implant post to ensure a point load application on the bottom of the abutment. Under these deliberately unfavorable conditions, screw self-loosening takes place with reasonable external load magnitudes, and thus the methodology can be experimentally validated.

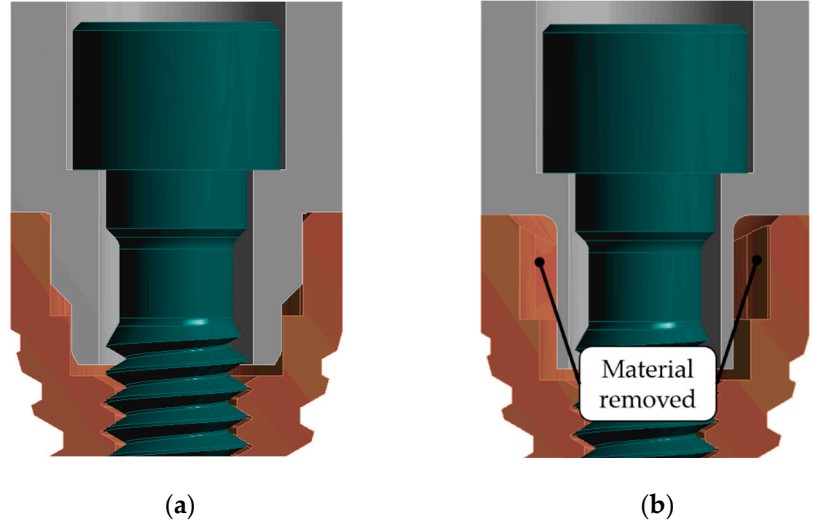

(**a**)     (**b**)

**Figure 6.** Implant abutment connection: (**a**) original, (**b**) modified.

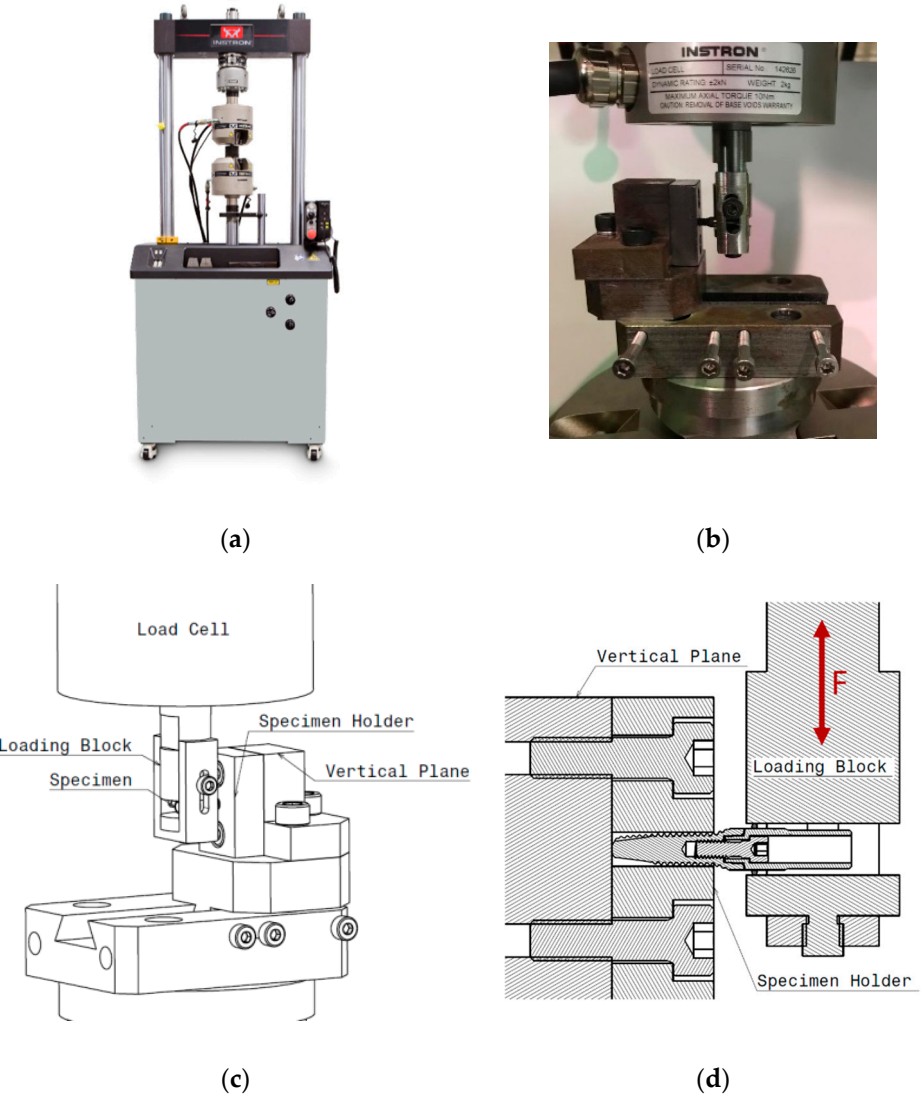

(**a**)     (**b**)

(**c**)     (**d**)

**Figure 7.** Experimental test: (**a**) test bench, (**b**) detail of the designed loading jig for tests, (**c**) drawing of the loading jig, (**d**) section view.

In order to validate the methodology under specific test conditions, three different screw tightening torques $T_T$ were applied: 10, 15 and 20 Ncm by means of a TOHNICHI BTG60CN-S torque gauge (2% accuracy). According to the methodology summarized in Figure 3, screw self-loosening under transverse loading occurs when $T_L \leq 0$ in Equation (8). Experimentally, loosening is usually detected and quantified by measuring the change of force in the screw [38]. Unfortunately, screws in dental implants are very small and hardly accessible, so the option of installing strain gauges was discarded. As a well-established alternative in dental implant literature, untightening torque after a number of external load cycles can be measured, since torque is proportional to preload according to Equation (3) [9,28,29,31–33].

Thus, 30 tightening and untightening operations were performed before testing (without applying external load cycles between both operations) in one of each preload cases in order to set the mean value of the loosening torque $T_{Li}$ referred to as initial untightening torque from now on and its standard deviation $\sigma$, inherent to the tightening operation [2]. Additionally, few tests were conducted to evaluate the load loss due to creep-relaxation with time; although this phenomenon is not expected to happen with metals at room temperature.

Then, the cyclic load was applied to the tested dental implant. Preliminary results indicate that, for the particular implant tested in this work, when self-loosening occurs, 1000 cycles are enough to promote a significant loosening. Thus, in each experimental test, the final untightening torque $T_{Lf}$ was measured after applying 1000 load cycles, and compared with $T_{Li}$. Self-loosening was considered to take place if $T_{Lf} < T_{Li} - 2\sigma$. The $2\sigma$ is used to account for the normal distribution [2] noting that the probability of fulfilling such criterion without occurring self-loosening is only 2.1%. This procedure was repeated in the 30 tests for each of the three initial tightening torques. Furthermore, in order to obtain experimentally the critical external load value that causes self-loosening $F_{exp}$ and compare it with the theoretical one $F_{crit}$, the staircase method was used [39]. This procedure is widely used in fatigue testing to obtain the value of the fatigue limit. Accordingly, an initial arbitrary external load $F$ is applied in the first test; if the screw self-loosens, the external load is decreased by 5N; if it does not, the external load is increased by 5N.

## 2.5. Finite Element Model

A finite element (FE) model of the implant under study is needed in step 3 of the methodology (see Figure 3) in order to achieve the axial load $F_a$ and the transverse load $F_e$ acting on the screw head. The authors used Ansys Workbench® 19 R1 software (Ansys Iberia S.L, Madrid, Spain) to run the analyses, which consist of two load steps: in the first one, the preload calculated in step 2 of the methodology was applied to the screw via pretension section. As explained, Equation (3) was used to calculate the preload $F_p$ for each of the three tightening torques applied, namely 232 N the corresponding preload for 10 Ncm, 349 N for 15 Ncm and 465 N for 20 Ncm. In the second load step, the external load $F$ was applied to the abutment through the ring shown in Figure 7 in order to reproduce the experimental conditions described in the previous section. Implant and abutment are made of grade 4 commercially pure titanium (Ti CP4) and the prosthetic screw is made of Ti6Al4V ELI (Ti Gr5). Chemical composition of both materials is described in Table 1. Both materials were modeled as linear elastic, with $E = 103$ GPa, $\nu = 0.35$ for CP4 and $\nu = 0.31$ for GR5.

The sole purpose of the FE analysis is to obtain the values of the loads transmitted to the screw, and not to reproduce the exact condition of screw self-loosening itself. Consequently, a relatively simplified FE model is used to minimize the complexity and cost of the analysis. Along this line, only half dental implant assembly was modeled, and accordingly half the preload and the external load were applied to the model. Cylindrical threads where modelled in the screw since the error introduced is negligible as demonstrated in authors' previous work [36]. Since stress results are not needed at all in the screw-loosening prediction methodology presented in this paper, a very refined mesh is not necessary. In this sense, a mesh sensitivity study was performed and the mesh shown in Figure 8 was finally chosen in order to have a good compromise between the accuracy of the results

(this is, loads acting on the screw) and the analysis cost. Quadratic elements were used with a total of 185.320 Degrees of Freedom (DoF). Target and contact elements were used to model the interface of the mating surfaces with a friction coefficient obtained experimentally as already pointed out. The external surface of the implant part attached to the specimen holder was held by a fixed support in the analyses. Finally, the forces transmitted to the screw head were directly obtained by using the force reaction tool of Ansys Workbench® to be used to test the self-loosening state using Equation (8).

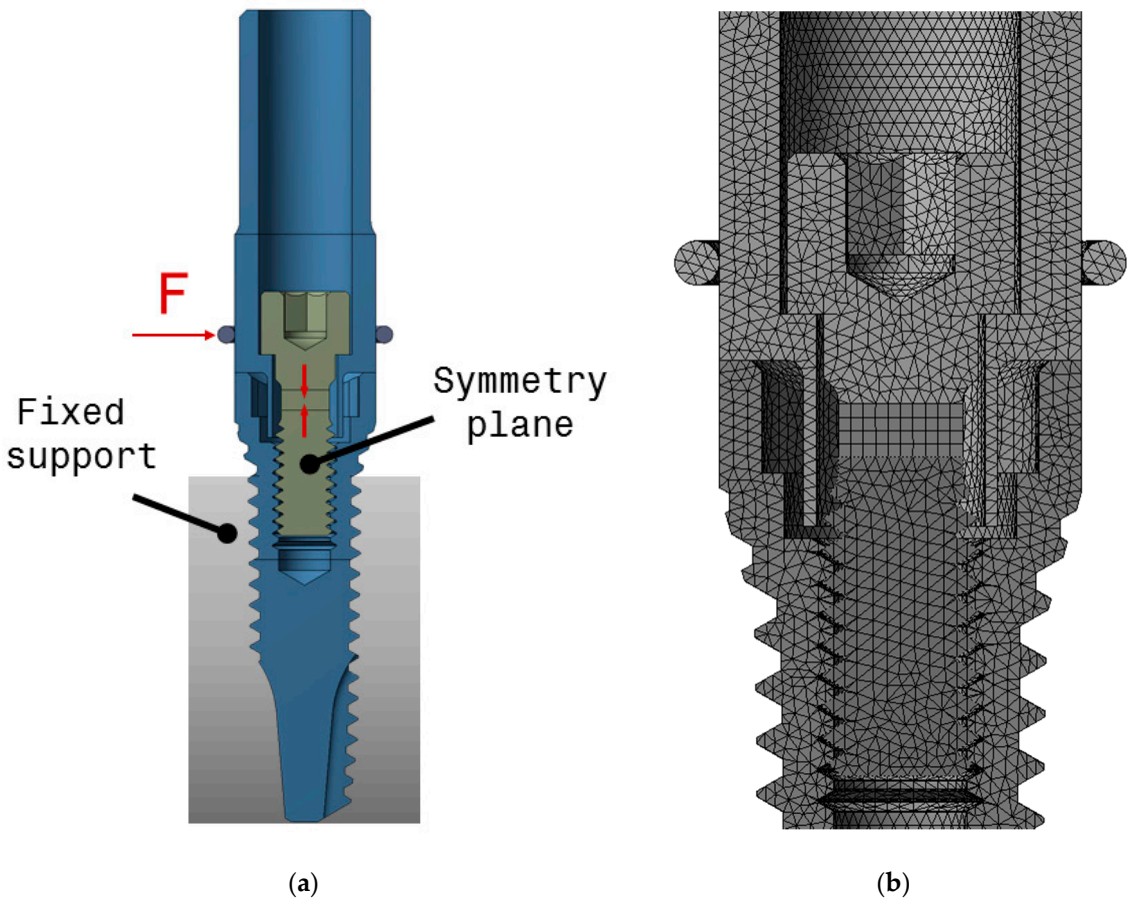

**(a)**                                    **(b)**

**Figure 8.** (**a**) Boundary conditions of the finite element (FE) model of the dental implant under study; (**b**) mesh.

**Table 1.** Chemical composition of materials used in implant and prosthetic component manufacturing process.

| Ti 6Al 4V ELI (TI GR5) | | TI CP4 | |
|---|---|---|---|
| **Composition** | **Wt.%** | **Composition** | **Wt.%** |
| Al | 5.5–6.5 | N(max) | 0.05 |
| V | 3.5–4.5 | C(max) | 0.08 |
| Fe(max) | 0.25 | Fe(max) | 0.5 |
| O(max) | 0.13 | O(max) | 0.4 |
| C(max) | 0.08 | H(max) | 0.0125 |
| N(max) | 0.05 | - | - |
| H(max) | 0.012 | - | - |

## 3. Results and Discussion

The new methodology was used to predict screw self-loosening under external cyclic loading for the dental implant under study, and the experimental tests were carried out to support the simplified

model and verify the results. Thus, the critical external load $F_{crit}$ for which screw self-loosening occurs was evaluated following the methodology in Figure 3: an external load $F$ was applied in the FE model, from this analysis $F_a$ and $F_e$ force reactions were obtained in the screw head and using the developed Equation (8) $T_L$ loosening torque value was achieved. The procedure was repeated iteratively by increasing the value of $F$ until $T_L$ falls to zero. As mentioned, three different tightening torques (10, 15 and 20 Ncm) were studied, and the obtained $F_{crit}$, $F_a$ and $F_e$ values for which self-loosening occur are shown in Table 2.

**Table 2.** Calculated load values that cause self-loosening in the dental implant under study using the methodology.

| $T_T$ (Ncm) | $F_{crit}$ (N) | $F_a$ (N) | $F_e$ (N) |
|---|---|---|---|
| 10 | 64.5 | 217.7 | 36.1 |
| 15 | 96.7 | 326.1 | 53.7 |
| 20 | 128.3 | 433.9 | 72.6 |

Then, the next step was to experimentally verify the values of the theoretically evaluated external forces $F_{crit}$ that cause self-loosening for the three initial tightening torques. The mean values of $T_{Li}$ obtained from measurements were 7.45, 11.6 and 16.0 Ncm with standard deviations of 0.4, 0.5 and 0.6 Ncm for the 10, 15 and 20 Ncm tightening torques, respectively. Figure 9 shows the scatter in untightening torques measured for each tightening torque.

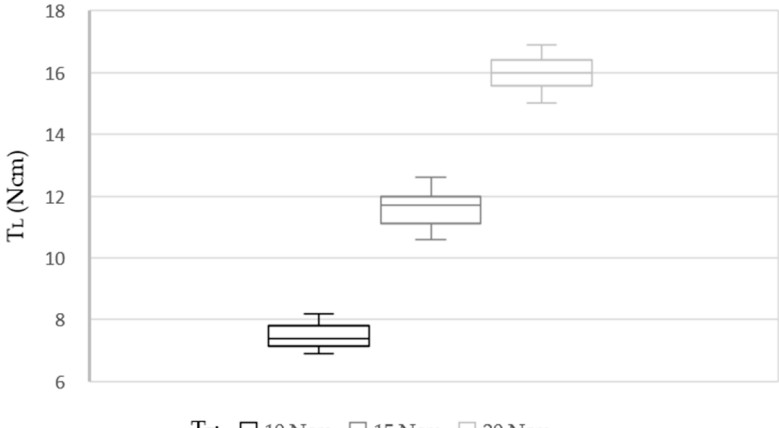

**Figure 9.** Box and whisker plot of all untightening operations performed before cyclic loading.

Then, 1000 cycles were applied in the experiment using the external loads close to the theoretical values $F_{crit}$ shown in Table 2. At the end of the 1000 cycles, $T_{Lf}$ was measured and compared with $T_{Li}$ through the $2\sigma$ criterion mentioned. Figure 10 shows the experimental results: tests where self-loosening occurred are marked with an 'x' symbol, while 'o' marks the tests with no loosening. Following the staircase method, the external load to cause self-loosening in the experimental setup ($F_{exp}$) is calculated as the average of the values indicated with black markers in Figure 10 while the grey markers are discarded. Table 3 summarizes the values of $F_{exp}$, their corresponding standard deviations $\sigma_{exp}$ and the difference between the experimental ($F_{exp}$) and theoretical ($F_{crit}$) results.

Furthermore, it has been appreciated that in the case under study, the presented methodology shows a constant relation between the external load at which self-loosening occurs and the tightening torque $\frac{F_{crit}}{T_T}$ as it can be deduced from Table 2. Experimental results were used to validate this statement, calculating the parameter $\frac{F}{T_T}$ for all the tests performed. Figure 10 shows not only the external load at which self-loosening occurred in each experimental test (primary axis) but also the parameter $\frac{F}{T_T}$ calculated for each test (secondary axis). All the experimental results, except the ones discarded

(grey markers in Figure 10), were used in an ANOVA to verify if the mentioned parameter $\frac{F}{T_T}$ can be considered constant. The data obtained from the ANOVA is shown in Table 4. According to the results obtained, the null hypothesis may be accepted (*p*-value = 0.28) so the fact that the mean values are the same cannot be rejected. Thus, $\frac{F}{T_T}$ can be assumed constant for this particular case, as predicted by methodology.

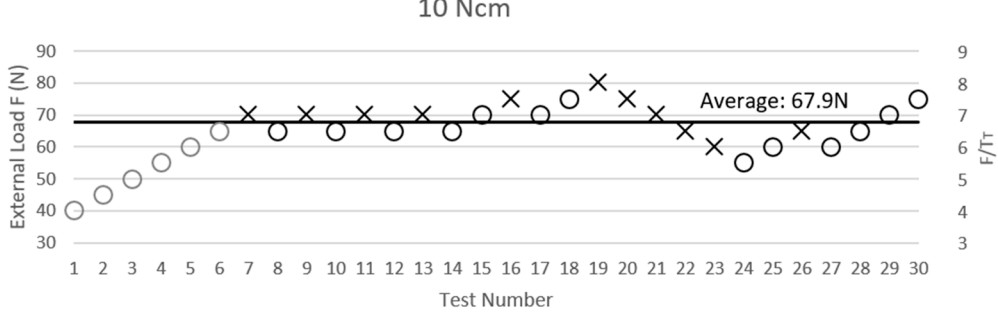

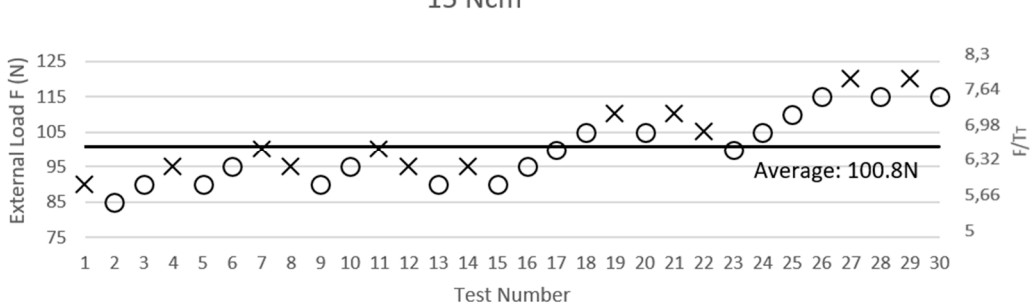

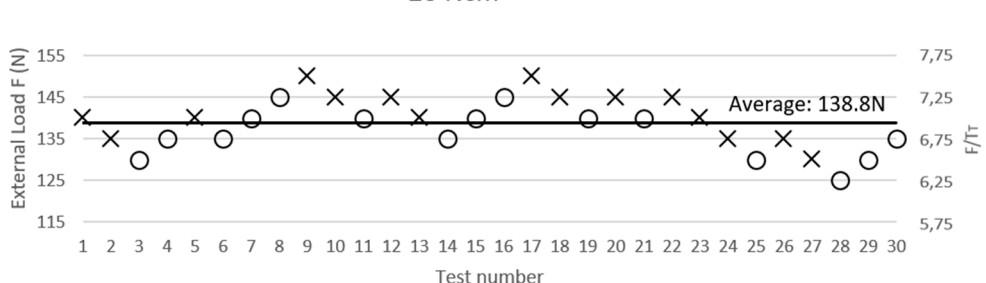

**Figure 10.** Calculation of $F_{exp}$ using the staircase method under 10, 15 and 20 Ncm tightening torques (left vertical axis) and $\frac{F_{exp}}{T_T}$ coefficient of each test (right vertical axis) ×: loosened; o: non loosened; grey marker: discarded value.

**Table 3.** External load values that cause self-loosening in the dental implant under study: methodology vs. experimental.

| $T_T$ (*Ncm*) | $F_{exp}$(*N*) | $\sigma_{exp}$(*N*) | Error (%) |
|---|---|---|---|
| 10 | 67.9 | 5.9 | 5 |
| 15 | 100.8 | 9.9 | 4.1 |
| 20 | 138.9 | 6.2 | 7.6 |

**Table 4.** ANOVA Table.

| Source of Variation | SS | DoF | MS | F | *p*-Value | $F_{crit}$ |
|---|---|---|---|---|---|---|
| Between groups | 0.75 | 2 | 0.38 | 1.29 | 0.28 | 3.11 |
| Within groups | 23.48 | 81 | 0.29 | | | |
| Total | 24.23 | 83 | | | | |

The good correlation between the theoretical $F_{crit}$ and experimental $F_{exp}$ results validate the proposed simplified approach and gives confidence in the presented methodology. Moreover, the theoretical results obtained by using the simplified approach $F_{crit}$ were compared with the results of the complex model of Nassar et al. [31,32] obtaining a difference smaller than 1% in all the cases studied in this work.

The proposed methodology is able not only to predict self-loosening under certain working conditions but it can also be used to understand how each variable affects the screwed joint behavior against self-loosening. From Equation (8) it can be deducted that a high preload is recommended to avoid screw loosening. This conclusion agrees with [25,40]. Friction coefficients of the screw head and thread contacts $\mu_h$ and $\mu_t$ are also demonstrated to highly affect the behavior against self-loosening. Thus, for a given preload, a high friction coefficient is recommended to avoid screw loosening according to the presented analytical tool and concurring with [21,22]. There are some other works where friction coefficient is reduced in the screwed joint contact by means of a coating or lubrication obtaining a better behavior against self-loosening [41,42]. This seems to disagree with the last statement. Nevertheless, in these works different friction coefficients are studied for the same tightening torque, rather than same preload. Hence, the results are not comparable since friction coefficient also affects the preload obtained [26,43]. Evidently, a higher tightening torque will lead to a higher preload and, therefore, to a better response against self-loosening.

The magnitude of the transverse load supported by the screw head also plays an important role in self-loosening [6,9,16,24]. From (8) it can be concluded that a robust design where the screw head suffers the less transverse load possible is desired since the transverse slippage is the cause of rotation, preload loss and, therefore, the screw loosening.

Finally, thread parameters affect the behavior of screwed joints against self-loosening as well. On one hand, a small thread pitch helps to avoid self-loosening. According to Nassar et al., screws with coarse threads would require a smaller loosening torque than those with fine threads [32]. This statement also agrees with [33,44]. On the other hand, the half angle of the thread profile $\alpha$ is recommended to be as high as possible according to [32]. From Equation (8), it can be concluded that a high angle of thread profile will lead to a higher frictional force in the threads, improving the screw-loosening response.

## 4. Conclusions

A simple analytical approach to assess the screw loosening condition under masticatory loading was developed based on the torque-preload formulation. The screw axial and transverse loads needed as input data are obtained from a simplified finite element analysis of the dental implant. The experimentally validated methodology can be used to predict self-loosening under given working conditions or even obtain the maximum value of load that a particular assembly can assume before self-loosening occurs. In this sense, the methodology is an extremely useful design tool for dental implant manufacturers and designers to select the appropriate geometry, thread configuration, friction coefficient values, or screw tightening torque in order to minimize screw self-loosening problems and consequently guarantee long-term stability and clinical success of dental implant fixation.

In addition, Equation (8) of the methodology contains all the critical design and operational parameters affecting self-loosening phenomenon in dental implants, and consequently some relevant clinical implications can be directly derived from it. Thus, the axial load in the screw is required to be as high as possible if a good self-loosening behavior is pursued; for such purpose, a high tightening

torque is recommended. Similarly, for a given screw preload force value, larger friction coefficient and effective contact radii values improve the response of the dental implant assembly. Finally, structurally robust implant-abutment connections will reduce the transverse load that reaches the screw and consequently screw self-loosening problems; this suggests that dental implants with wide platforms or Morse taper implant-abutment connections are less prone to suffer self-loosening, as future research will aim to study.

**Supplementary Materials:** The following are available online at http://www.mdpi.com/2076-3417/10/19/6748/s1. Supplementary Material: Development of the expression (8) of the manuscript.

**Author Contributions:** Conceptualization, M.A. (Mikel Abasolo); methodology, M.A. (Mikel Armentia); software, I.C.; validation, M.A. (Mikel Armentia); formal analysis, M.A. (Mikel Abasolo); investigation, A.-H.B.; resources, M.A. (Mikel Abasolo); data curation, I.C.; writing—original draft preparation, M.A. (Mikel Armentia); writing—review and editing, M.A. (Mikel Armentia), A.-H.B.; visualization, I.C.; supervision, I.C.; project administration, M.A. (Mikel Abasolo); funding acquisition, M.A. (Mikel Abasolo) All authors have read and agreed to the published version of the manuscript.

**Funding:** This work has received financial support of the Basque Government [grant number IT947-16].

**Acknowledgments:** The authors are very grateful to Raúl Cosgaya for the valuable assistance provided in the experimental tests performed for this work.

**Conflicts of Interest:** The authors declare no conflict of interest.

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
