# Peer review of "On the Use of a Simplified Slip Limit Equation to Predict Screw Self-Loosening of Dental Implants Subjected to External Cycling Loading"

_applsci, doi:10.3390/app10196748_

Round 1

Reviewer 1 Report

Armentia and co-authors report their novel approach to predict self-loosening of dental implant.

The manuscript’s incredibly long and the lengthy equations/explanation dampens reader’s interest from the very beginning. All these details could very well be suited for the supplementary section. Throughout the manuscript, the authors fail to show data in a simple box-whisker plot (mean and standard deviation).

Apart from several errors (typos, references, etc.), the authors write statements presuming that the reviewers are aware of their previous studies. This is definitely not the case. There are several instances where the authors do this, repeatedly. Please rectify.

The manuscript title indicates cyclic loading; however, there is no instance of any cyclic testing in the data.

Why is no data shown for the Pin-on-disk tribometer? What is the use of showing the setup, if the authors don’t report data?

Figure 8 – The differences are barely recognized here. The authors should provide an appropriate image with labels to assist the reader. This image does not contribute to the manuscript in anyway.

Figure 10 – Do all the parts have same material properties? What is the contact relationship between (i) support and base, and (ii) dental screw and Details of the materials and their respective properties utilized in the FEA should be outlined separately in a table.

Figure 11 - Does not make sense. Why do the authors use such plot way of showing their data? What is the use of plotting TT vs TL, instead of showing individual implant performance data? Please use box and whisker plots with each data point labelled.

For the mechanical testing, how were the loading levels of 10, 15, and 20 Ncm chosen? What is the rationale behind using these levels? Are these levels reflected in the finite element model? 5Ncm difference between the experimental conditions is not a viable study design. I understand that the manufacturer’s specification are detailed at 30Ncm, but how does this fit in the realm of the study?

Figure 12 - What is the meaning of this plot? Other than poorly plotted data points, this figure does not contribute anything to the manuscript.

I would recommend the authors to perform a statistical test to see if there are differences in the data obtained. Based on the appropriate statistical results only, the authors can make the assumption that their implant is actually “loosening”, or there is not difference between the experimental settings?

There are lot more errors that I can possibly list here.

Reviewer 2 Report

Abstract: Some numerical results would be helpful.

Dropped line at 46?

Generally a good introduction. Some further discussion on the impact of different materials and surface finishes would be helpful.

Please make it clear which equations are taken from the literature and which have been derived here.

An estimation of the maximum error associated with the difference in the Fe/Fa curves is required.

Error- reference source not found? - Line 202

Fig 6 - scale bar?

Figure 7 - some arrows to describe what is being shown would be helpful

At lines 247-250 - please include the range in which these were varied.

How was the implant held in the vertical plane?

The description of the mechanical tests lacked some clarity and could be improved.

Further description of the FEA model is required - mesh convergence tests, boundary conditions etc

307, 322, 335 - Error!

Why were the grey markers discarded?

The paper has no discussion section? This is completely unacceptable. Without this, essentially this is an experimental report and cannot be published. The authors need to consider the trends, conclusions, compare it with other work and summarise. This cannot be achieved within 2 paragraphs of conclusion.

Round 2

Reviewer 1 Report

Regardless of the simplification of the FE mesh, mesh sensitivity should be reported. The convergence of the stress field with respect to mesh resolution is not clear: please provide the norm, the convergence measure quantity and threshold which allow to replicate your computational model setup.

Additionally, there are no details of calculations from FEA. Please provide (i) a detailed figure of stress changes before and after forces are applied on the implant, and (ii) how and where the changes in stress were considered relevant to the study (since there are lot of uneven structures in the model). 

Fig 4 - Typo in the text

Line 188 - what type of cyclic loading? Please elaborate on the details of the test. How many steps? What was the frequency? What was the loading strain? Was pre-strain applied? How was the data recorded? What data output resulted from the test? What are the deviations over the cycles? It is the task of the authors to make their papers relevant and not the reviewer's task to dictate the necessary details. 

This is unacceptable "Further details about this procedure will be given in the Results and discussion section." Materials and Methods should be detailed in their respective sections and elaborate them separately. There should be no overlap between these sections. 

Reviewer 2 Report

Thank you for addressing my concerns.

Round 3

Reviewer 1 Report

No further comments. 

This manuscript is a resubmission of an earlier submission. The following is a list of the peer review reports and author responses from that submission.

Round 1

Reviewer 1 Report

Thank you for for the opportunity of reading this manuscript.

I have read the paper and found it to be an interesting and well-performed engineering study of a methodology to study screw self-loosening phenomenon in preliminary design stages of dental implants.

While the work provides a solid foundation for the design of a methodology applicable in preliminary design stages of dental implants, it does not seem to be of interest to the dentist but rather to the manufacturer. In my opinion, this work does not have the general suitability for the journal special issue.

I would emphasize that this decision is not a reflection of the quality of work, but rather of its field of relevance and interest.

Reviewer 2 Report

Introduction- too long

Please put references at the end of the sentence- lines 54-55, 69, 98-9,103.

In some sentences is written “  The more accurate expression  for (2) is given in [32]”

I understand that it means equation 2 as given in reference 34 but the authors have to give the info from the reference itself.

The mathematical part is long and hard to follow for dental practitioner Fig 5 gives a flowchart to calculate the self-loosening force however, you need to find some variables such as -friction coefficient – it is hard to find or calculate.

In the dental literature we use Ncm and not Nmm

You wrote that 30 tightening and untightening operations were performed before testing. What was the torque controller type, did you calibrate it each time? To what torque? How this effect screw threads that was not designed for this?

Please add more details about the implant macro-structure,  parallel wall?  Pitch, lead,  

Reviewer 3 Report

The submitted article gives a very interesting idea of analysis of dental implant strength of materials.
The work simplifies the models to a simple equation for use in a step-by-step 80 methodology to predict screw self-loosening of dental implants. It is interesting research for the dental prosthesis producers. However, it would be beneficial if a link between the technical aspect of the result was made to some medical consequences. I mean the dental tools or some technical surgical recomendations.